# Anti-Tumor Effect of Heat-Killed *Bifidobacterium bifidum* on Human Gastric Cancer through Akt-p53-Dependent Mitochondrial Apoptosis in Xenograft Models

**DOI:** 10.3390/ijms23179788

**Published:** 2022-08-29

**Authors:** SukJin Kim, Hwan Hee Lee, Wonhyeok Choi, Chang-Ho Kang, Gun-Hee Kim, Hyosun Cho

**Affiliations:** 1Department of Bio-Health Convergence Major, Duksung Women’s University, Seoul 01369, Korea; 2Department of Pharmacy, Duksung Women’s University, Seoul 01369, Korea; 3Duksung Innovative Drug Center, Duksung Women’s University, Seoul 01369, Korea; 4Mediogen Co., Ltd., Jecheon 27159, Korea

**Keywords:** gastric cancer, paraprobiotics, xenograft, apoptosis

## Abstract

Paraprobiotics, inactivated microbial cells, regulate immune system and exhibit antioxidant and anti-inflammatory activities in patients with weakened immunity or the elderly. This study evaluated the anti-tumor effects of heat-killed *Bifidobacterium* and *Lactobacillus* on human gastric cancer MKN1 cells in vitro and in vivo in xenograft animal models. First, cytotoxicity and apoptosis in MKN1 cells of 11 different heat-killed *Bifidobacterium* or *Lactobacillus* strains were examined using the MTT assay or flow cytometry, respectively. Then, BALB/c nude mice xenograft animal models were implanted with human gastric cancer MKN1 cells and orally administered a selected single or a mixture of heat-killed bacterial strains to investigate their inhibitory effect on tumor growth. In addition, the expression of p-Akt, p53, Bax, Bak, cleaved caspase-9, -3, and PARP in the tumor tissues was analyzed using Western blotting assay or immunohistochemistry staining. The results show that heat-killed *B. bifidum* MG731 (MG731), *L. reuteri* MG5346 (MG5346), and *L. rhamnosus* MG5200 (MG5200) induced relatively greater apoptosis than other strains in MKN1 cells. Oral administration of a single dose or a mixture of MG731, MG5346, or MG5200 significantly delayed tumor growth, and MG731 had the most effective anti-tumor effect in the xenograft model. Protein expression of p-Akt, p53, Bax, cleaved caspase-3 and -9, and PARP in tumors derived from the xenograft model correlated with the results of the immunohistochemistry staining.

## 1. Introduction

Paraprobiotics that confer health benefits to humans and animals are also known as non-viable microbial cells, crude cell extracts, ghosts, or inactivated probiotics [1,2]. Heat-killed *Bifidobacterium* (*B.*) and *Lactobacillus* (*L.*) are representative strains of paraprobiotics that have been reported to exert anti-inflammatory, antibacterial, and immunomodulatory effects [3,4,5]. Recently, as interest in the application of paraprobiotics as pharmaceuticals has increased, research on paraprobiotics for cancer prevention and treatment has continued [6]. In a recent animal study, *B. longum* induced anti-tumor activity by reducing liver tumor size and arresting the cancer cell cycle [7]. Heat-killed *L. brevis*, *L. plantarum*, and *L. paracasei* significantly suppress proliferation and induce apoptosis in breast cancer and colorectal cancer cells [8,9]. Paraprobiotics are among the main drug candidates to replace surgery, radiation therapy, and chemotherapy for gastric cancer prevention and treatment [10].

Gastric cancer (GC) is the third most deadly cancer in the world, with 1.09 million reported cases and an estimated 769,000 deaths in 2020, according to the World Health Organization’s 2022 report [11]. Current chemotherapy drugs for cancer, such as cispla-tin, docetaxel, and carboplatin, have various undesirable side effects, including mouth sores, fatigue, nausea, vomiting, and diarrhea [12]. Heat-killed *L. brevis* or *L. paracasei* have an inhibitory effect on the growth of human colon cancer HT-29 cells by increasing apoptosis [9]. Cell-free supernatants of *B. bifidum* inhibit the growth of colon cancer cells and are associated with substantial improvement in gastrointestinal cancer [13,14]. It has been reported that short-chain fatty acids (SCFA), the main component of paraprobiotics, has various advantages for the host, such as the improvement of intestinal barrier function, glucose homeostasis, and immune regulation, and is particularly involved in the activation and differentiation of immune cells [15,16]. Moreover, SCFA exhibits antiproliferative, apoptotic, and differentiation properties of cancer cells through induction of histone hyperacetylation [17,18]. Thus, paraprobiotics differentiate between normal cells and cancer cells and induce apoptosis in cancer cells [19,20].

Apoptosis, programmed cell death, is induced through the activation of several signaling molecules and caspases in cancer cells [21]. Apoptosis is initiated through the PI3K/Akt signaling pathway. Phosphatidylinositol 3-kinase (PI3K) is activated by stimulated tyrosine kinase growth factor receptors to induce Akt activation [22]. Akt, a serine/threonine kinase, is overexpressed in carcinomas and regulates cell survival processes, such as proliferation, differentiation, and apoptosis. It has been reported that Akt phosphorylates Mdm2, a ubiquitin ligase, to regulate p53 activity and interferes with p53-mediated pro-apoptosis transcriptional reaction [23]. p53, the most representative tumor suppressor gene, induces apoptosis via the intrinsic mitochondria-mediated apoptotic pathway. p53 exerts anti-tumor effects by increasing the expression of pro-apoptotic proteins, such as Bax, Bak, and caspase-9, through endogenous stimuli, such as oxidative stress, DNA damage, and ischemia [24]. The release of cytochrome c from the inner part of the mitochondrial membrane into the cytoplasm leads to cleavage and activation of the caspase cascade. Caspase-9, an initiator caspase, is activated by the mitochondrial apoptosis factor and induces the cleavage of caspase-3 and poly ADP-ribose polymerase (PARP), which are the executive and terminator caspases, respectively [25,26]. 

In the present study, we aimed to investigate the anti-tumor effects of heat-killed *Bifidobacterium* and *Lactobacillus* against human gastric cancer. We screened the cytotoxicity and apoptotic activity of 11 heat-killed *Bifidobacterium* or *Lactobacillus* strains against human gastric cancer MKN1 cells. In addition, we investigated the anti-tumor effect of selected heat-killed *Bifidobacterium* and *Lactobacillus* strains in xenograft animal models implanted with human gastric cancer MKN1 cells. Moreover, we attempted to elucidate the molecular mechanism underlying the anti-tumor activity of heat-killed *Bifidobacterium* and *Lactobacillus*.

## 2. Results

### 2.1. Cytotoxic Effect of Heat-Killed Bifidobacterium or Lactobacillus on Human Gastric Cancer MKN1 Cells

To investigate the cytotoxic effect of heat-killed *Bifidobacterium* or *Lactobacillus* bacterial strains on human gastric cancer MKN1 cells, the cells were treated with 1 × 10^8^ or 1 × 10^9^ cells/mL of heat-killed *Bifidobacterium* or *Lactobacillus*, respectively, for 24 h. Cytotoxicity was determined using the MTT assay, and morphological changes were observed under a microscope (Figure 1). The viability of MKN1 cells was significantly decreased by 1 × 10^9^ cells/mL of 11 bacterial strains, which was correlated with distinguished observations under microscopy (Figure 1a,b). Furthermore, 1 × 10^9^ cells/mL of *B. bifidum* MG731 (25.77%), *L. reuteri* MG5346 (57.33%), and *L. rhamnosus* MG5200 (56.84%) showed the highest cytotoxic effect on MKN1 cells. 

### 2.2. Apoptotic Effect of Heat-Killed Bifidobacterium or Lactobacillus on Human Gastric Cancer MKN1 Cells

We examined the effect of heat-killed *Bifidobacterium* and *Lactobacillus* on the apoptosis of gastric cancer MKN1 cells using Annexin V-FITC and PI staining. As shown in Figure 2, the percentage of cells analyzed by flow cytometry was classified into four categories: early apoptosis (Annexin V+/PI− quadrants), late apoptosis (Annexin V+/PI+ quadrants), living cells (Annexin V−/PI− quadrants), and dead cells (Annexin V−/PI+ quadrants). The total apoptosis (%) was defined as the sum of early apoptosis (%) and late apoptosis (%) (Figure 2b). MKN1 cells were treated with or without 1 × 10^9^ cells/mL heat-killed *Bifidobacterium* or *Lactobacillus* for 24 h, and apoptosis was analyzed by flow cytometry. The cells treated with 11 bacterial strains each showed a significantly higher apoptosis effect (%) than control cells (4.64%) (*p* < 0.05). The 1 × 10^9^ cells/mL of *B. bifidum* MG731, *L. reuteri* MG5346, *L. rhamnosus* MG316, and *L. rhamnosus* MG5200 dramatically increased the induction of apoptosis by 13.26%, 24.63%, 13.40%, and 11.41%, respectively. 

### 2.3. Anti-Tumor Effects of Heat-Killed Bifidobacterium or Lactobacillus in Xenograft Models Bearing Human Gastric Cancer MKN1 Cells

The anti-tumor effect of selected heat-killed *Bifidobacterium* and *Lactobacillus* was evaluated using a xenograft animal model bearing human gastric cancer MKN1 cells. All in vivo data were generated by six mice per group. Figure 3a,b shows that all treatments significantly inhibited tumor growth compared with drinking water on day 19. Notably, the tumor volume was 318 ± 15 mm^3^ in the control group, 136 ± 22 mm^3^ in the MG731 group, 184 ± 12 mm^3^ in the MG5346 group, 187 ± 9 mm^3^ in the MG5200 group, 177 ± 24 mm^3^ in the 3Mix-1 group, and 122 ± 27 mm^3^ in the 3Mix-3 group (Figure 3b). Interestingly, the tumor volume in MG731 and 3Mix-3 groups was dramatically reduced by 43% and 38%, respectively, compared with the control group (*p* < 0.05). The MG731, MG5200, and 3Mix-3 groups showed a significant tumor growth inhibitory effect compared with the control group from the 7th day, earlier than the other groups. Tumor weight was significantly reduced in the MG731 (39 ± 11 mg), MG5346 (56 ± 8 mg), MG5200 (45 ± 16 mg), 3Mix-1 (44 ± 9 mg), and 3Mix-3 (36 ± 6 mg) groups compared with the control group (78 ± 12 mg). Thus, tumor weight positively correlated with tumor volume (Figure 3c). 

There was no significant difference between the experimental groups and the control group during the experimental period, indicating that there was no toxicity caused by the oral administration of heat-killed *Bifidobacterium* or *Lactobacillus* (Figure 3d).

### 2.4. Apoptotic Effects of Heat-Killed Bifidobacterium or Lactobacillus via the Akt-p53 Signaling Pathway in MKN1 Cell-Derived Tumors

We investigated the expression of apoptotic signaling molecules in tumor tissues derived from xenograft models implanted with human gastric cancer MKN1 cells (Figure 4 and Figure 5). The Akt-p53 signaling pathway regulates cancer cell proliferation and apoptosis by activating downstream transcription factors such as Bax and Bak, which promote apoptosis in cancer cells. As shown in Figure 4a,b, the expression of p53 was increased in the tumor tissues of all treatment groups, especially in 3Mix-3, which was increased by 2.48-fold compared with that in the control (Figure 4b). The expression of p-Akt/Akt in tumor tissues derived from all treatment groups was significantly lower than that in the control (*p* < 0.05). The expression levels of Bax/GAPDH were significantly increased in the MG731, MG5200, and 3Mix-3 groups compared with that in the control (Figure 4d). Furthermore, Bak/GAPDH expression levels were increased by 1.05, 1.86, 1.17, and 2.49 times in the MG5346, MG5200, 3Mix-1, and 3Mix-3 groups, respectively, compared with the control (Figure 4e). 

We confirmed the expression of apoptotic signaling molecules in tumor tissues derived from the xenograft models. Caspase-9, an initiator caspase, initiates apoptosis signal transduction, and caspase-3 and PARP, executive caspases, perform bulk proteolysis to induce apoptosis [26]. The expression of cleaved caspase-9 was significantly higher in tumor tissues derived from the MG731, MG5200, and 3Mix-3 groups than in the control group. (Figure 5a,b). Compared with the control group, the MG731 and 3Mix-3 groups increased by 2.30- and 2.09-fold, respectively. The expression level of cleaved caspase-3 was increased by more than 1.5-fold in the experimental group compared with the control group and was the highest in the MG731 and 3Mix-3 groups (Figure 5a–c). The expression level of cleaved PARP increased in all treatment groups compared with that in the control group, and the 3Mix-3 group showed the highest increase (Figure 5a–d).

Using IHC staining, we confirmed the increase in the expression levels of p-Akt, p53, Bax, cleaved caspase-9 and -3, and PARP in the tumor tissues derived from MKN1 cell-implanted xenograft mice that were orally administered a mixture of *B. bifidum* MG731, *L. reuteri* MG5346, *L. rhamnosus* MG5200, or a mixture of *B. bifidum* MG731, *L. reuteri* MG5346, and *L. rhamnosus* MG5200 compared with the others (Figure 6).

## 3. Discussion

In the present study, we demonstrated the anti-tumor effect of heat-killed *Bifidobacterium* and *Lactobacillus* bacterial strains on human gastric cancer MKN1 cells in vitro and in vivo in a xenograft model. In several in vivo studies, a combined administration of *B. bifidum* MG731 and PD-1 showed a strong anti-tumor effect in a colorectal cancer mouse model, and a mixture of heat-killed *B. bifidum* MG731, *L. casei* MG4584, and *L. reuteri* MG5346 had a synergistic effect to delay tumor growth in a xenograft model implanted with human colorectal cancer RKO cells [19,27]. Although various studies have reported the anti-tumor effects of live *Bifidobacterium* and *Lactobacillus*, there have been few in vivo studies on gastric cancer; the anti-tumor effects in xenograft models implanted with human gastric cancer cells are still unclear.

In previous studies, 1 × 10^9^ cells of probiotics *B. bifidum*, *L. casei*, *L. reuteri*, and *L. rhamnosus* were orally administered in clinical trials, especially in cancer patients for 4–12 weeks [28,29]. Therefore, we used heat-killed bacterial strains at concentrations of 1 × 10^8^–1 × 10^9^ cells to evaluate human gastric cancer MKN1 cell cytotoxicity. As shown in Figure 1, we observed a high cytotoxic effect of heat-killed bacterial strains at 1 × 10^9^ cells/mL in gastric cancer MKN1 cells. In several studies, cell free supernatant of *B. bifidum* extract had the highest cytotoxicity with 50% among *B. adolescentis*, *B. animalis subsp. lactis*, *B. animalis subsp. animalis*, *B. bifidum*, and *B. angulatum* in human colorectal cancer HT-29 and caco-2 cells [30,31]. In a previous in vitro study, heat-killed *L. plantarum* KU15149, *L. brevis* KU15159, and *L. brevis* KU15176 had approximately 35% cytotoxicity against the human stomach adenocarcinoma (AGS) gastric cancer cell line [32]. Similarly, in our study, the cytotoxicity of *Lactobacillus* to human gastric cancer cells averaged 36.5% and was the highest in *L. reuteri* MG5346 and *L. rhamnosus* MG5200 (Figure 1b). Moreover, the cytotoxicity of *B. bifidum* MG731 in MKN1 cells was more than 75%. MKN1 cells were used in this study because it is easier to observe anti-tumor activity through AKT-related apoptosis than other gastric cancer cells such as AGS, Hs746T, and LMSU [33]. According to previous studies, *Lactobacillus* and *Bifidobacterium* exhibited similar anti-tumor effects in several gastric cancer cell lines such as MKN-1, MKN-28, AGS, SNU1, SNU216, SNU-601, SNUC2A, NIH/3T3, and Jurkat. Therefore, the bacterial strains used in this study are speculated to show high anti-tumor effects in other gastric cancer cells [34,35].

Increasing cancer cell apoptosis prevents tumor invasiveness, angiogenesis, cell proliferation, and the accumulation of mutations that interfere with differentiation [36]. We demonstrated that 1 × 10^9^ cells/mL of all heat-killed bacterial strains significantly increased total apoptosis compared with the control (Figure 2b). It has been reported that probiotic *Lactobacillus* species exhibit excellent total apoptosis in AGS cancer cells through Akt and Bax signaling [37]. Therefore, we speculated that the superior total apoptosis of *B. bifidum* MG731, *L. reuteri* MG5346, *L. rhamnosus* MG316, and *L. rhamnosus* MG5200 occurs through Akt and Bax signaling mechanisms. There was no precise correlation between cytotoxicity and apoptotic effects of the heat-killed bacterial strains tested in our study (Figure 1 and Figure 2). We speculate that cancer cell death is triggered not only by apoptosis but also by other signaling pathways [38,39]. In addition, different methods for measuring cytotoxic activity and apoptosis contribute to the numerical variability of the results. In fact, in our previous study, we used the same paraprobiotic bacterial strains and demonstrated excellent anti-tumor effects of *B. bifidum* MG731, *L. casei* MG4584, and *L. reuteri* MG5346 with high cancer cell cytotoxicity and apoptosis in a xenograft model implanted with human colorectal cancer cells [19].

Therefore, we selected heat-killed *B. bifidum* MG731, *L. reuteri* MG5346, and *L. rhamnosus* MG5200 for an in vivo xenograft model based on the above in vitro study as well as the companies’ capability of mass production of the bacterial strains. As shown in Figure 3, oral administration of a single or a mixture of selected heat-killed bacterial strains significantly inhibited tumor growth in the gastric cancer xenograft model. The MG5346, MG5200, and 3Mix-1 groups inhibited tumor growth by more than 41.20%, and the MG731 and 3Mix-3 groups significantly suppressed tumor growth by approximately 57.32% compared with the control. MG731 showed the highest tumor growth inhibitory effect among the three strains, which is consistent with the results of the in vitro experiment. A recent report that probiotic-derived molecules exhibit tumor growth inhibitory effects in a xenograft model partly agrees with our in vivo results [40,41]. In a previous study, the combined administration of *B. bifidum* MG731, *L. casei* MG4584, and *L. reuteri* MG5346 showed anti-tumor effects on colorectal cancer by synergistic effect, whereas in this study, the single administration of *B. bifidum* MG731 showed excellent anti-tumor effects in gastric cancer [19].

Subsequently, we examined the anti-tumor mechanisms of the selected heat-killed bacterial strains in our xenograft models. According to previous in vitro studies, the cell-free culture supernatant of *Lactiplantibacillus plantarum* exhibited cytotoxicity and apoptosis activity through regulation of Bax, caspase-3, and caspase-9 in gastric cancer AGS cells [37,42]. We identified a more holistic mechanism of apoptosis and found that heat-killed *B. bifidum* MG731, *L. reuteri* MG5346, and *L. rhamnosus* MG5200 regulated the expression of p-Akt, p53, Bax, and Bak in xenograft-derived tumor tissues (Figure 4 and Figure 6). With regards to the process of mitochondrial apoptosis, cleavage of caspase by p53-bax signaling induces apoptosis and eventually leads to cancer cell death [24]. We confirmed that heat-killed *B. bifidum* MG731, *L. reuteri* MG5346, and *L. rhamnosus* MG5200 increased the expression of caspase-9, and -3 and PARP in the tumor tissues (Figure 5 and Figure 6). In many in vitro studies, *B. bifidum* and *L. rhamnosus* extracts have been shown to induce apoptosis in human cancer cells through mitochondrial cytochrome c release and cleavage of caspase-9 and -3 [28,43]. The results of the current study (Figure 5) are consistent with these previous studies. The MG731 and 3Mix-3 groups, which showed the fastest tumor growth inhibitory effect, mostly regulated the expression of apoptotic proteins. We demonstrated that the tumor growth inhibitory activities of *B. bifidum* MG731, *L. reuteri* MG5346, and *L. rhamnosus* MG5200 were caused by Akt-p53-dependent apoptosis. 

Paraprobiotics include cell wall components, such as (lipo)teichoic acids, cell surface-associated proteins, neutral and acidic polysaccharides, peptidoglycan, and proteinaceous filaments [44]. Another recent in vitro study reported that the peptidoglycan of *L. paracase* has anti-tumor activity by inducing caspase-3 activation and that lipoteichoic acid and 5-FU of *Bifidobacterium* significantly inhibit tumor proliferation by inducing apoptosis [45,46]. Another in vivo study reported clinical and animal model validation that a mixture of *Lactobacillus* and *Bifidobacterium* could significantly lower postoperative inflammation, enhance immunity, resume gut microbiota composition, and promote postoperative recovery [47]. According to previous studies, paraprobiotic bacterial strains including *B. bifdum* MG731, *L. casei* MG311, *L. rhamnosus* MG316, *S. thermophilus* MG5140, etc., had no cytotoxic to RAW264.7 cells, and especially *B. bifdum* MG731, which had the highest SCFA content among 17 bacterial strains, has high antioxidant activity and anti-inflammatory effects, showing the potential of functional paraprobiotics [48,49]. We speculated that the anti-tumor effects of heat-killed MG731 exhibit gastric cancer growth inhibitory effects via molecules involved in SCFAs, lipoteichoic acid, mannoprotein, and fimbriae. However, further studies are needed to elucidate the composition of *B. bifidum* MG731 and its efficacy in immunity or cancer.

## 4. Materials and Methods

### 4.1. Preparation of Bacterial Strains 

The powder of the heat-killed bacterial strains used in this study was provided by MEDIOGEN Co., Ltd. (Jecheon, Korea). The bacterial cultures were centrifuged at 5000× *g* for 5 min to obtain a cell pellet. The cell pellet was heat-killed at 100 °C for 30 min and freeze-dried. The bacterial strains were dissolved in cell medium and drinking water for in vitro and in vivo experiments, respectively. The origin and species of the 11 bacterial strains used in this study are shown in Table 1.

### 4.2. Cell Culture

Human gastric cancer MKN1 cells were purchased from the Korean Cell Line Bank (Seoul, Korea). MKN1 cells were cultured in Roswell Park Memorial Institute Medium 1640 (RPMI 1640; Corning Inc., New York, NY, USA) supplemented with 1% penicillin/streptomycin (Gibco, Grand Island, NY, USA) and 10% inactivated fetal bovine serum (FBS; Young-In Frontier, Seoul, Korea). Cells were incubated at 37 °C in a humidified atmosphere containing 5% CO_2_.

### 4.3. Cell Cytotoxicity

Cytotoxicity of heat-killed bacterial strains against human gastric cancer MKN1 cells was detected using the MTT assay. MKN1 cells were plated at a density of 5 × 10^3^ cells/well in 96-well plates and were incubated for 24 h. Following incubation, cells were incubated with heat-killed bacterial strains (1 × 10^8^ or 1 × 10^9^ cells/mL) for 24 h. After incubation, the cells were treated with the MTT solution (1 mg/mL) for 4 h. The supernatant was removed and 100 μL of dimethyl sulfoxide (DMSO; Sigma-Aldrich, St. Louis, MO, USA) was added to each well to dissolve formazan crystals. Absorbance was assessed using a microplate reader (BMG Labtech, Offenburg, Germany) at 450 nm.

### 4.4. Cell Morphology

The morphology of human gastric cancer MKN1 cells was observed using light microscopy (×200) (Nikon Eclipse TS100, Nikon, Tokyo, Japan). MKN1 cells were seeded at a density of 1 × 10^6^ cells/well in 6-well plates. After incubation for 24 h, the cells were treated with heat-killed bacterial strains (1 × 10^9^ cells/mL) for 24 h. The cells were observed under the light microscope.

### 4.5. Annexin V-FITC/Propidium Iodide Staining Assay

Apoptosis was evaluated using the Annexin V-FITC Apoptosis Detection Kit I (Roche, Basel, Switzerland). Briefly, MKN1 cells were cultured at a density of 1 × 10^6^ cells in 6-well culture plates for 24 h. Following incubation, the cells were treated with heat-killed *Bifidobacterium* or *Lactobacillus* (1 × 10^9^ cells/mL) for 24 h. Cells were collected and stained with Annexin V-FITC and propidium iodide (PI) for 15 min in the dark. Cell apoptosis was immediately analyzed using a flow cytometer (EasyCyte Guava, Merck Millipore). 

### 4.6. Human Gastric Cancer Xenografts in BALB/c Nude Mice

The animal experiments were approved by the Institutional Animal Care and Use Committee of Duksung Women’s University (permit number:2021-011-015). Five-week-old female BALB/c nude mice were purchased from Raonbio (Seoul, Korea). Mice were housed in accommodated in a pathogen-free controlled environment (45 ± 5% humidity and 23–27 °C under a 12-h day/12-h night cycle) and acclimatized for one week before experiments. Mice were fed standard laboratory chow and water ad libitum. Human gastric cancer MKN1 cells (5 × 10^6^ cells/mouse) were subcutaneously injected into the back, next to the right hind leg. Heat-killed bacterial strains dissolved in drinking water were orally administered daily for 19 d. Mice were randomly assigned to the following six groups (*n* = 6/group); (1) control group: drinking water; (2) MG731 group: *B. bifidum* MG731 (1 × 10^9^ cells/mouse), 1 × 10^9^ cells/mouse based on in vitro studies; (3) MG5346 group: *L. reuteri* MG5346 (1 × 10^9^ cells/mouse); (4) MG5200 group: *L. rhamnosus* MG5200 (1 × 10^9^ cells/mouse); (5) 3 Mix-1 group: 1:1:1 mixture of *B. bifidum* MG731, *L. reuteri* MG5346, and *L. rhamnosus* MG5200 (1 × 10^9^ cells/mouse); and (6) 3 Mix-3 group: 1:1:1 mixture of *B. bifidum* MG731, *L. reuteri* MG5346, and *L. rhamnosus* MG5200 (3 × 10^9^ cells/mouse). Tumor volume was measured daily using a standard caliper and calculated as follows: tumor volume = (tumor length (mm) × tumor width (mm)^2^)/2 [50,51]. After oral administration, tumors were harvested and weighed.

### 4.7. Western Blotting

Apoptosis-related protein expression in tumor tissues was examined using Western blotting analysis [50]. The specimens were lysed using extraction buffer (Intron Biotechnology, Seoul, Korea) to extract proteins, which were quantified by Bradford (Coomassie) Protein Assay (GenDEPOT, Katy, TX, USA). Proteins were separated by electrophoresis and transferred to polyvinylidene fluoride (PVFD) microporous membranes (Merck Millipore, Burlington, MA, USA). Membranes were incubated with the primary and secondary antibodies. Anti-Akt (#9272), anti-p-Akt (#4060), anti-p53 (#2527), anti-Bax (#2774), anti-Bak (#21205), apoptosis Sampler Kit (#9915), anti-GAPDH (#5174), anti-rabbit IgG (#7074), and anti-mouse IgG (#7076) were obtained from Cell Signaling Technology (CST; Danvers, MA, USA) and used for the incubation. Proteins were visualized using chemiluminescence detection (ECL System, Bio-Rad, Hercules, CA, USA) and quantified using the Image J program (National Institutes of Health, Bethesda, MD, USA). 

### 4.8. Immunohistochemistry (IHC)

IHC staining of tumor tissues was performed according to a previously described standard method [50]. Tumor tissue blocks were prepared using a Frozen Section Compound (Leica, Hesse, Germany) and sectioned to a thickness of 4 μM. The tissues were blotted with primary antibodies in PBST overnight at 4 °C. Anti-p-Akt (#4060), anti-p53 (#2527), anti-Bax (#2774), anti-cleaved caspase-3 (#9664), and anti-cleaved PARP (#5625) were purchased from CST, and anti-cleaved caspase-9 (#SN-40504) was obtained from Signalway Antibody (College Park, MD, USA). Following treatment, the cells were treated with a secondary antibody, mouse anti-rabbit IgG-HRP (#sc-2537, Santa Cruz, Dallas, TX, USA), for 2 h at room temperature. The slices were then stained with DAB (Vector Laboratories, Burlingame, CA, USA) and visualized under a microscope (×200 and ×400). 

### 4.9. Statistical Analysis

All in vitro data were analyzed in triplicate and are presented as the mean ± standard deviation (SD). In vivo data were analyzed from 6 animals per group. Statistical analyses were performed using one-way analysis of variance (ANOVA) with Duncan’s multiple range test. Data were analyzed using SPSS version 22 (IBM Corp., Armonk, NY, USA). *p* < 0.05 was regarded as statistically significant.

## 5. Conclusions

Our study demonstrated that heat-killed *Bifidobacterium* and *Lactobacillus* induced apoptosis of human gastric cancer MKN1 cells in vitro, and heat-killed *B. bifidum* MG731 had outstanding anti-tumor effects in an MKN1 cell-implanted xenograft mouse model through p53 signaling, including p-Akt, p53, Bax, Bak, caspase-9 and -3, and PARP. Our findings suggest the possibility of using heat-killed *B. bifidum* MG731 for the treatment of gastrointestinal cancer because it has excellent anti-tumor effects on both colorectal and gastric cancers.

## Figures and Tables

**Figure 1 ijms-23-09788-f001:**
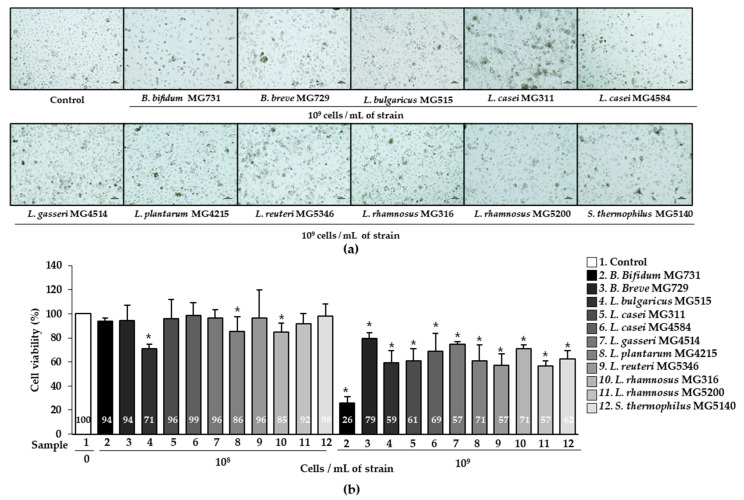
Cytotoxicity of heat-killed *Bifidobacterium* and *Lactobacillus* in human gastric cancer MKN1 cells analyzed by MTT assay. (**a**) Cell morphology and (**b**) cell viability (%). Results are presented as mean ± SD in three independent experiments. * *p* < 0.05 compared with the control.

**Figure 2 ijms-23-09788-f002:**
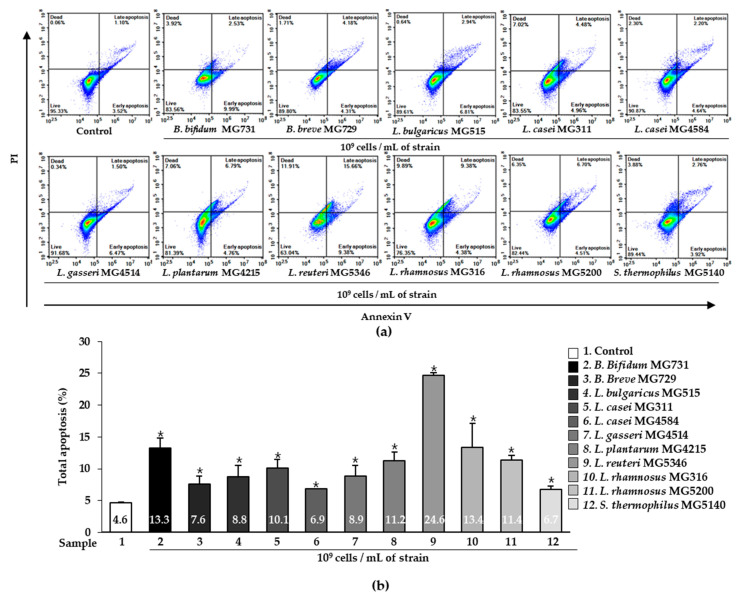
Apoptosis of heat-killed *Bifidobacterium* and *Lactobacillus* in human gastric cancer MKN1 cells analyzed by flow cytometry assay. (**a**) Representative Annexin V and PI staining for apoptosis and (**b**) total apoptosis (%) in MKN1 cells. Results are presented as mean ± SD in three independent experiments. * *p* < 0.05 compared with the control.

**Figure 3 ijms-23-09788-f003:**
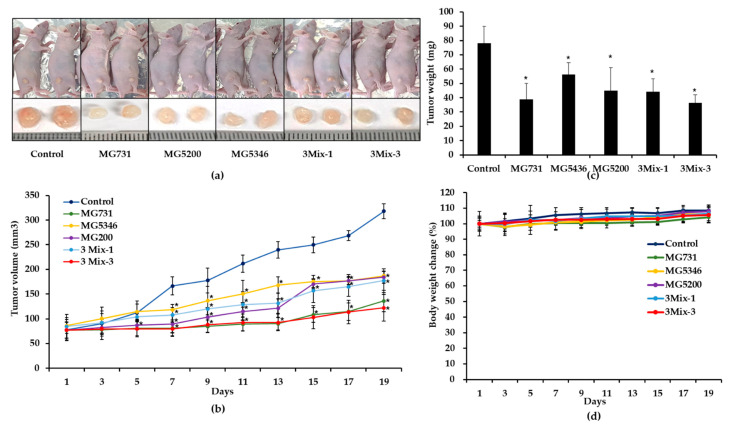
The tumor growth inhibitory effect of heat- killed *Bifidobacterium* and *Lactobacillus* in xenograft model implanted with human gastric cancer MKN1 cells. BALB/c nude mice were subcutaneously injected with 5 × 10^6^ MKN1 cells/mouse into the dorsum next to right hind leg. The mice were orally administrated drinking water or a single or mixed strain of heat-killed bacterial strains every day for another 19 days. (**a**) Photographs represent tumor size of xenograft model mice. (**b**) Tumor growth curve. (**c**) Tumor weight (mg). (**d**) Body weight changes. Results are presented as the mean ± SD (*n* = 6). * *p* < 0.05 compared with the control group. Control, drinking water; MG731, *B. bifidum* MG731 (1 × 10^9^ cells/mouse); MG5346, *L. reuteri* MG5346 (1 × 10^9^ cells/mouse); MG4584, *L. rhamnosus* MG5200 (1 × 10^9^ cells/mouse); 3Mix-1, 1:1:1 mixture of *B. bifidum* MG731 + *L. reuteri* MG5346 + *L. rhamnosus* MG5200 (1 × 10^9^ cells/mouse); 3Mix-3, mixture of 3Mix-3, 1:1:1 mixture of *B. bifidum* MG731 + *L. reuteri* MG5346 + *L. rhamnosus* MG5200 (3 × 10^9^ cells/mouse).

**Figure 4 ijms-23-09788-f004:**
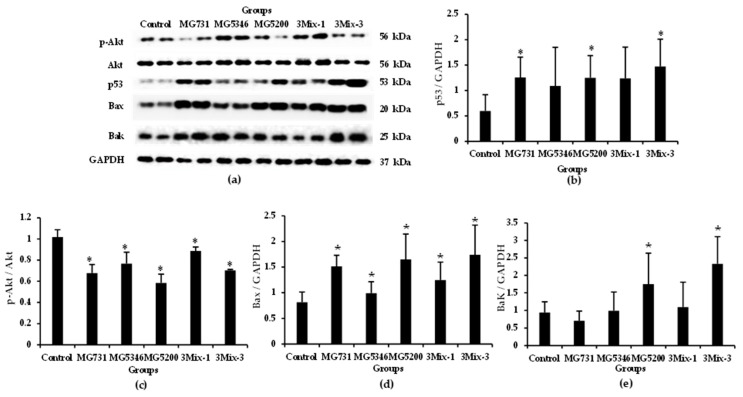
Expression of apoptosis-related proteins in tumor tissues from xenograft model analyzed by Western blotting. After 19 days of administration, proteins were extracted from the tumor tissues and subjected to Western blotting. (**a**) The representative Western blot images; Western blot quantitative data of (**b**) p-Akt/Akt, (**c**) p53/GAPDH, (**d**) Bax/GAPDH, and (**e**) Bak/GAPDH. Results are presented as the mean ± SD (*n* = 6). * *p* < 0.05 compared with the control group.

**Figure 5 ijms-23-09788-f005:**
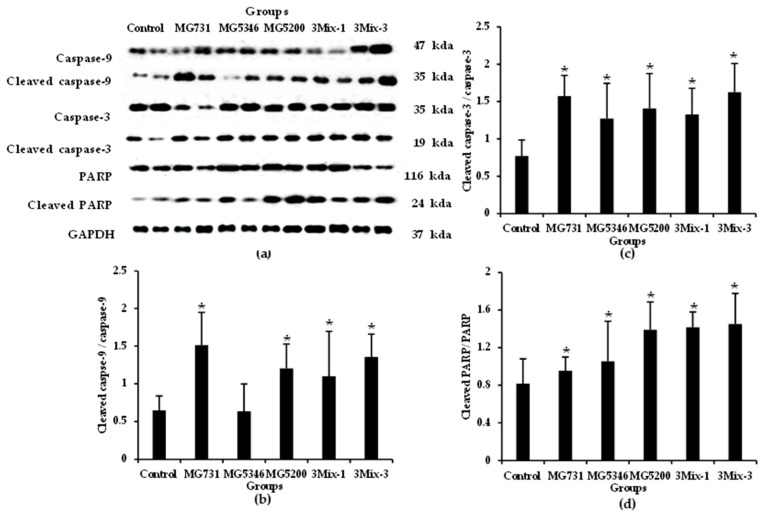
Expression of apoptosis-related proteins in tumor tissues from xenograft model analyzed by Western blotting assay. After 19 days of heat-killed single or mixed strain treatment, proteins were extracted from the tumor tissues and subjected to Western blotting. (**a**) The representative Western blot images; Western blot quantitative data of (**b**) cleaved caspase-9/caspase-9, (**c**) cleaved caspase-3/caspase-3, and (**d**) cleaved PARP/PARP. Results are presented as the mean ± SD (*n* = 6). * *p* < 0.05 compared with the control group.

**Figure 6 ijms-23-09788-f006:**
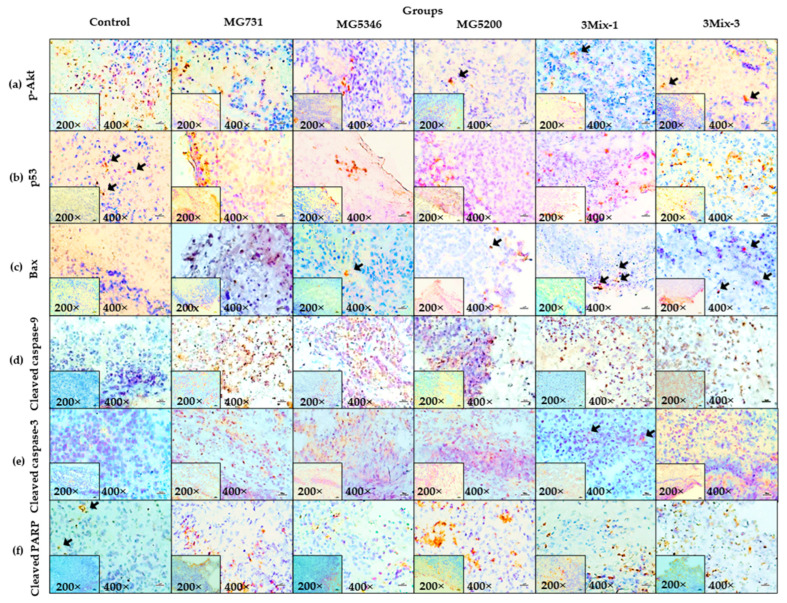
Immunohistochemistry analysis of the tumor tissues. After 19 days of heat-killed single or mixed strain treatment, tumor tissues were harvested and stained by (**a**) p-Akt, (**b**) p53, (**c**) Bax, (**d**) cleaved caspase-9, (**e**) cleaved caspase-3, and (**f**) cleaved PARP. The scale bar represents 100 px. Brown dots are expression of p-Akt, p53, Bax, cleaved caspase-9 and -3, PARP. Black arrows indicate the brown dots of IHC (magnification 200× or 400× as indicated).

**Table 1 ijms-23-09788-t001:** The origins and species of the 11 bacterial strains.

Origin	Species	Strain
Infant feces	*B. bifidum*	MG731
	*B. breve*	MG729
	*L. rhamnosus*	MG316
Fermented food	*S. thermophiles*	MG5140
	*L. bulgaricus*	MG515
	*L. casei*	MG311
	*L. rhamnosus*	MG5200
Human	*L. casei*	MG4584
	*L. gasseri*	MG4514
	*L. plantarum*	MG4215
Food	*L. reuteri*	MG5346

## Data Availability

Not applicable.

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
