# Peer review of "Anti-Tumor Effect of Heat-Killed Bifidobacterium bifidum on Human Gastric Cancer through Akt-p53-Dependent Mitochondrial Apoptosis in Xenograft Models"

_ijms, 2022, doi:10.3390/ijms23179788_

Round 1
Reviewer 1 Report
Revision of ijms-1862522
Within their manuscript “Anti-tumor effect of heat-killed Bifidobacterium bifidum MG731 on human gastric cancer through Akt-p53 dependent mitochondrial apoptosis in xenograft models”, the authors examine the cytotoxic effect of different heat-inactivated bacteria on the gastric cancer cell line MKN1 and mouse xenografts. Exposure to bacteria significantly reduced the viability of cancer cells in vitro and in vivo, as shown by a xenograft mouse model. This study might be of relevance for applied sciences and future treatment strategies and – thus – represents and interesting field in cancer research in general. However, there appear to be some major concerns of the experimental setup, data analysis and presentation, which are listed in detail below.
11. There is a capital letter in the surname of the first author.
22. An additional sentence at the beginning of the abstract to provide more background information would be beneficial for the reader to better understand the aim of the study.
33. Line 42,43: “Paraprobiotics are among the main drug candidates to replace surgery, radiation therapy, and chemotherapy for cancer prevention and treatment.” The authors should provide at least one reference for this statement.
44. The authors should check the citation of reference 9, as it is once stated for breast cancer and once for colon cancer.
55. The authors should additionally cite https://doi.org/10.1016/j.phanu.2020.100219.
66. Line 51,52: “Thus, paraprobiotics differentiate between normal cells and cancer cells and induce apoptosis in cancer cells. “ It would be interesting to provide some information why parabiotics target cancer cells, but not healthy cells. Please add additional information.
77. Overall, the work would significantly benefit from analyzing a second gastric cancer cell line. As different cell lines often display great behavioral differences, it is necessary to check for cytotoxic effects also in other GC cell lines. Additionally, considering the statement of the authors (point 3), that parabiotics distinguish between cancer and non-cancer cells, the authors should also analyze a non-cancer cell line to clarify the effect of the heat-killed bacteria on healthy cells, either separately or in a mixed culture cultivated with cancer cells. Without those experiments, it is hard to make a statement how effective this procedure is in general and to which extend it specifically affects cancer cells vs healthy cells.
88. Figure 1: I highly encourage the authors to use a different labelling for Figure 1b, as the shades of grey are very hard to distinguish. Labelling the bars and the figure legends with e.g. “1”, “2”,… would simplify the interpretation.
99. Figure 1b: I encourage the authors to increase their n numbers, as the statistical analysis from 3 values might not be completely trustworthy. Hence, the statistics do not seem to match the presented data, i.e. a “*” significance for e.g. B. bifidum MG731 compared to the control seems incomprehensible considering the small error bar. Please provide the values of the individual measurements additionally to the bars and clarify the results obtained within the discussion.
110. Figure 2b: While the authors stated within the methods that they measured their samples in triplicates, it is not clear if n=3 means triplicates of the same sample or three different samples were measured. It is also not stated in the methods, how the authors define n=3.
111. Figure 3b,c,d: The authors MUST extend their description of the figures and method section. The description of Figure 3c is not sufficient to understand the data presented. Where all 8 tumors analyzed within Figure 3b and c or just some? Where all tumors analyzed in multiple measurements? Why was n=5 analyzed in Figure 3d and what happened to the other 3 mice per group (considering the authors´ statements that 8 mice per group were monitored)?
112. Figure 3c: Please change the units for uniformity – while mg in stated in the axis, g is stated in the legend.
113. Reference to Figure 4a and 4b is missing in the text.
114. Figure 4a: It is unclear if the same sample was loaded twice or two different samples were loaded for the western blots. Was the same mouse tested in triplicates or were three samples form different mice analyzed? The authors stated that 8 animals were part of each group – where all 8 animals tested? The authors must clarify within the methods and the figure legends how many samples were analyzed (and used for the statistics) and if n=3 describes the triple analysis of the 8 animals or that only 3 animals were analyzed. If only 3 animals were analyzed, the authors should show and re-analyze the data of all animals.
115. Figure 4a: Considering the differences of lane 11 and 12 (3Mix-3), is seems that different samples were analyzed, as lane 12 displays more p53 with the same amount of protein loaded. Did the authors find any correlation with the tumor volume within these 2 samples? Similarly within lane 9 and 10 (3Mix-1), lane 9 seems to display a more prominent p53 band compared to lane 10, although there seems to be slightly more protein loaded within lane 10. Besides physiological differences of live animals, one would assume a different tumor mass of this samples. Did the authors observe any correlation of the amount of e.g. p53 and the tumor volume? The authors should re-analyze their data accordingly and add a respective figure/panel showing any correlation.
116. Figure 4 b-e: In line with point 8, again, showing the individual values within the graphs would simplify the interpretation for the reader throughout all figures.
117. The authors might re-name Figure 4c to Figure 4b and Figure 4b to Figure 4c.
118. Figure 5a: Is it correct that the same samples as shown in Figure 4a were analyzed within this blot? The amount of pro-apoptotic markers in lane 12 seems to be (again) more prominent compared to lane 11. In line with point 10, more data and additional analysis are necessary.
119. The size (in kDa) of the proteins analyzed within the western blots should be added for comprehensibility.
220. Figure 6: The number of arrows within the images seems to be set voluntarily to visually match the western blot data. Can the authors describe how the position of the arrows was chosen? There are many dots that are not marked by arrows and arrows that do not seem to point to positive signals. The authors should remove all arrows, as they do not represent the actual results from the images. This would omit the false presentation of e.g. 3 arrows for p53 in MG5346, although the signals obtained is comparable to MG731 (where 7 arrows are shown). Interestingly, this arrow number “matches” the band intensity of the western blot data. Instead, the authors should show 2 representative arrows per image: one (e.g. in red) that displays the minimal signal that was counted as positive and one arrow (e.g. in blue) that points towards very clear signal. This would simplify the interpretation of the results. Additionally, I encourage the authors to re-analyze the images to determine the area of positive signal (in % of total tissue area shown, as it is hard to distinguish individual spots).

Author Response
Within their manuscript “Anti-tumor effect of heat-killed Bifidobacterium bifidum MG731 on human gastric cancer through Akt-p53 dependent mitochondrial apoptosis in xenograft models”, the authors examine the cytotoxic effect of different heat-inactivated bacteria on the gastric cancer cell line MKN1 and mouse xenografts. Exposure to bacteria significantly reduced the viability of cancer cells in vitro and in vivo, as shown by a xenograft mouse model. This study might be of relevance for applied sciences and future treatment strategies and – thus – represents and interesting field in cancer research in general. However, there appear to be some major concerns of the experimental setup, data analysis and presentation, which are listed in detail below.
- There is a capital letter in the surname of the first author.
Answer: In fact, the first author has published a couple of paper with name written as SukJin Kim, therefore, we want to keep the same way in this manuscript.
- An additional sentence at the beginning of the abstract to provide more background information would be beneficial for the reader to better understand the aim of the study.
Answer: We added more background information to the abstract.
Line 15-16: Paraprobiotics, inactivated microbial cells, regulate immune system and exhibit anti-inflammatory, and antioxidant activities in patients with weakened immunity or the elderly.
- Line 42,43: “Paraprobiotics are among the main drug candidates to replace surgery, radiation therapy, and chemotherapy for cancer prevention and treatment.” The authors should provide at least one reference for this statement.
Answer: We added a reference to the manuscript.
Line 43-44: Paraprobiotics are among the main drug candidates to replace surgery, radiation therapy, and chemotherapy for gastric cancer prevention and treatment [10].
- The authors should check the citation of reference 9, as it is once stated for breast cancer and once for colon cancer.
Answer: We revised the manuscript to fit the reference.
Line 41-43: Heat-killed L. brevis, L. plantarum, and L. paracasei significantly suppress proliferation and induce apoptosis in breast cancer and colorectal cancer cells [8, 9].
- The authors should additionally cite https://doi.org/10.1016/j.phanu.2020.100219.
Answer: We added references to the manuscript.
- Line 51,52: “Thus, paraprobiotics differentiate between normal cells and cancer cells and induce apoptosis in cancer cells.” It would be interesting to provide some information why parabiotics target cancer cells, but not healthy cells. Please add additional information.
Answer: We add additional information in the manuscript.
Line 53-59: It has been reported that short-chain fatty acids (SCFA), the main component of paraprobiotics, has various advantages for the host, such as improvement of intestinal barrier function, glucose homeostasis and immune regulation, and is particularly in-volved in the activation and differentiation of immune cells [15, 16]. Moreover, SCFA exhibits antiproliferative, apoptotic, and differentiation properties of cancer cells through induction of histone hyperacetylation [17, 18].
- Overall, the work would significantly benefit from analyzing a second gastric cancer cell line. As different cell lines often display great behavioral differences, it is necessary to check for cytotoxic effects also in other GC cell lines. Additionally, considering the statement of the authors (point 3), that parabiotics distinguish between cancer and non-cancer cells, the authors should also analyze a non-cancer cell line to clarify the effect of the heat-killed bacteria on healthy cells, either separately or in a mixed culture cultivated with cancer cells. Without those experiments, it is hard to make a statement how effective this procedure is in general and to which extend it specifically affects cancer cells vs healthy cells
Answer: Based on the reviewers' comments, we added information on the effects of several paraprobiotics on other gastric cancer cells and normal cells.
Line 253-259: MKN1 cells were used in this study because it is easier to observe antitumor activity through AKT related apoptosis than other gastric cancer cells such as AGS, Hs746T, LMSU [33]. According to previous studies, Lactobacillus and Bifidobacterium exhibited similar anti-tumor effects in several gastric cancer cell lines such as MKN-1, MKN-28, AGS, SNU1, SNU216, SNU-601, SNUC2A, NIH/3T3, and Jurkat. Therefore, the bacterial strains used in this study are speculated to show high anti-tumor effects in other gastric cancer cells [34, 35].
Line 316-320: According to previous studies, paraprobiotic bacterial strains including B. bifdum MG731, L. casei MG311, L. rhamnosus MG316 and S. thermophilus MG5140 etc. had no cytotoxic to RAW264.7 cells, and especially B. bifdum MG731, which had the highest SCFA content among 17 bacterial strains, has high antioxidant activity and anti-inflammatory effects, showing the potential of functional paraprobiotics [48, 49].
- Figure 1: I highly encourage the authors to use a different labelling for Figure 1b, as the shades of grey are very hard to distinguish. Labelling the bars and the figure legends with e.g. “1”, “2”,… would simplify the interpretation.
Answer: We added a different labelling and revised figure.
- Figure 1b: I encourage the authors to increase their n numbers, as the statistical analysis from 3 values might not be completely trustworthy. Hence, the statistics do not seem to match the presented data, i.e. a “*” significance for e.g. bifidum MG731 compared to the control seems incomprehensible considering the small error bar. Please provide the values of the individual measurements additionally to the bars and clarify the results obtained within the discussion.
Answer: We inserted individual measures into the graph and corrected the statistics.
- Figure 2b: While the authors stated within the methods that they measured their samples in triplicates, it is not clear if n=3 means triplicates of the same sample, or three different samples were measured. It is also not stated in the methods, how the authors define n=3.
Answer: We conducted three different experiments each and modified the sentences.
Line 99, 125 : Results are presented as mean ± SD in three independent experiments.
- Figure 3b,c,d: The authors MUST extend their description of the figures and method section. The description of Figure 3c is not sufficient to understand the data presented. Where all 8 tumors analyzed within Figure 3b and c or just some? Where all tumors analyzed in multiple measurements? Why was n=5 analyzed in Figure 3d and what happened to the other 3 mice per group (considering the authors´ statements that 8 mice per group were monitored)?
Answer: Of the 8 mice tested, 6 mice with normal tumor growth on average were selected per group and the data were used. Figures 3b,c,d used data from 6 tumors per group, respectively. We revised the incorrect part of the manuscript and added this part.
- Figure 3c: Please change the units for uniformity – while mg in stated in the axis, g is stated in the legend.
Answer: We revised the figure legend.
- Reference to Figure 4a and 4b is missing in the text.
Answer: We mentioned 4a and 4b in the sentence.
- Figure 4a: It is unclear if the same sample was loaded twice or two different samples were loaded for the western blots. Was the same mouse tested in triplicates or were three samples from different mice analyzed? The authors stated that 8 animals were part of each group – where all 8 animals tested? The authors must clarify within the methods and the figure legends how many samples were analyzed (and used for the statistics) and if n=3 describes the triple analysis of the 8 animals or that only 3 animals were analyzed. If only 3 animals were analyzed, the authors should show and re-analyze the data of all animals.
Answer: Of the 8 mice tested, 6 mice with normal tumor growth on average were selected per group to use the data. We performed western blotting of all 6 tumors per group once. We corrected the errors in the manuscript.
- Figure 4a: Considering the differences of lane 11 and 12 (3Mix-3), is seems that different samples were analyzed, as lane 12 displays more p53 with the same amount of protein loaded. Did the authors find any correlation with the tumor volume within these 2 samples? Similarly, within lane 9 and 10 (3Mix-1), lane 9 seems to display a more prominent p53 band compared to lane 10, although there seems to be slightly more protein loaded within lane 10. Besides physiological differences of live animals, one would assume a different tumor mass of this samples. Did the authors observe any correlation of the amount of e.g. p53 and the tumor volume? The authors should re-analyze their data accordingly and add a respective figure/panel showing any correlation.
Answer: Based on the reviewers' comments, we reanalyzed each tumor Western blot result.
- Figure 4 b-e: In line with point 8, again, showing the individual values within the graphs would simplify the interpretation for the reader throughout all figures.
Answer: According to the reviewer's opinion, the labeling per group on the x-axis of the graph was reconfirmed.
- The authors might re-name Figure 4c to Figure 4b and Figure 4b to Figure 4c.
Answer: Based on reviewer comments, we have renamed 4b and 4c.
- Figure 5a: Is it correct that the same samples as shown in Figure 4a were analyzed within this blot? The amount of pro-apoptotic markers in lane 12 seems to be (again) more prominent compared to lane 11. In line with point 10, more data and additional analysis are necessary.
Answer: The samples in Figure 5a and Figure 4a are identical, and we rearranged the entire western blotting image in the order of the samples used.
- The size (in kDa) of the proteins analyzed within the western blots should be added for comprehensibility.
Answer: We added size (in kDa) to the western blot images of Figures 4a and 5a.
- Figure 6: The number of arrows within the images seems to be set voluntarily to visually match the western blot data. Can the authors describe how the position of the arrows was chosen? There are many dots that are not marked by arrows and arrows that do not seem to point to positive signals. The authors should remove all arrows, as they do not represent the actual results from the images. This would omit the false presentation of e.g. 3 arrows for p53 in MG5346, although the signals obtained is comparable to MG731 (where 7 arrows are shown). Interestingly, this arrow number “matches” the band intensity of the western blot data. Instead, the authors should show 2 representative arrows per image: one (e.g. in red) that displays the minimal signal that was counted as positive and one arrow (e.g. in blue) that points towards very clear signal. This would simplify the interpretation of the results. Additionally, I encourage the authors to re-analyze the images to determine the area of positive signal (in % of total tissue area shown, as it is hard to distinguish individual spots).
Answer: Immunohistochemistry is an experimental method for detecting a target protein in a tissue, and the expression level of the protein cannot be observed. Therefore, it is difficult to mark a signal that was considered positive or a clear signal, as the reviewers mentioned. We remarked with black arrows only the spots with 5 or fewer stained points in the image.
Reviewer 2 Report
This study by Kim et al examines the effects of various bacterial cultures (in their case heat inactivated) on gastric cancer cells, both in vitro and in xenograft models. The experiments appear to have been carefully performed and show that the bacteria induce apoptosis through activation of the caspase cascade. Unfortunately, the study suffers from the drawback seen in most of these sorts of investigations in that no information is provided on what component of the bacteria actually induces the apoptosis. For all the authors know it could be a bacterial toxin, a protein carbohydrate, nucleic acid, lipid etc.. That said, My feeling is the manuscript should be published after the minor criticisms have been addressed.
The authors don't explain the bacterial culture which was added to the cells-was the pellet dissolved in water or suspended in water?? Was the supernatant used?
The western blots shown in Figure 5 are not of the highest quality-there is no GAPDH loading control. Is the very high level of caspase 9 in the final right hand track real? The authors should show the cleaved products (caspase 3, 9 and PARP) on the same gel as the intact protein. Also there is very large variability in the caspase 9 gels, comparing the 3 versions presented. In 1 of them the caspase 9 expression is low and in 1 very high. I don't find these blotting data satisfactory.
Author Response
This study by Kim et al examines the effects of various bacterial cultures (in their case heat inactivated) on gastric cancer cells, both in vitro and in xenograft models. The experiments appear to have been carefully performed and show that the bacteria induce apoptosis through activation of the caspase cascade.
- Unfortunately, the study suffers from the drawback seen in most of these sorts of investigations in that no information is provided on what component of the bacteria actually induces the apoptosis. For all the authors know it could be a bacterial toxin, a protein carbohydrate, nucleic acid, lipid etc. That said, My feeling is the manuscript should be published after the minor criticisms have been addressed.
Answer: Based on the reviewers' comments, we have added considerable amount of references to which components of the paraprobiotics induce cancer cell apoptosis.
Line 53-59: It has been reported that short-chain fatty acids (SCFA), the main component of paraprobiotics, has various advantages for the host, such as improvement of intestinal barrier function, glucose homeostasis and immune regulation, and is particularly in-volved in the activation and differentiation of immune cells [14, 15]. Moreover, SCFA exhibits antiproliferative, apoptotic, and differentiation properties of cancer cells through induction of histone hyperacetylation [16, 17].
Line 315-319: According to previous studies, paraprobiotic bacterial strains including B. bifdum MG731, L. casei MG311, L. rhamnosus MG316 and S. thermophilus MG5140 etc. had no cytotoxic to RAW264.7 cells, and especially B. bifdum MG731, which had the highest SCFA content among 17 bacterial strains, has high antioxidant activity and anti-inflammatory effects, showing the potential of functional paraprobiotics [44, 45].
- The authors don't explain the bacterial culture which was added to the cells-was the pellet dissolved in water or suspended in water?? Was the supernatant used?
Answer: The ‘Material & Method’ in manuscript was revised according to the reviewers' opinions.
Line 327-330: The bacterial cultures were centrifuged at 5,000 × g for 5 minutes to obtain a cell pellet. The cell pellet was heat-killed at 100°C for 30 minutes freeze-dried. The bacterial strains were dissolved in cell medium and drinking water for in vitro and in vivo experiments, respectively.
- The western blots shown in Figure 5 are not of the highest quality-there is no GAPDH loading control. Is the very high level of caspase 9 in the final right hand track real? The authors should show the cleaved products (caspase 3, 9 and PARP) on the same gel as the intact protein. Also there is very large variability in the caspase 9 gels, comparing the 3 versions presented. In 1 of them the caspase 9 expression is low and in 1 very high. I don't find these blotting data satisfactory.
Answer: We added GAPDH data according to the reviewer's comments. In the case of the Western blotting, the sizes of the proteins are very different, so it is impossible to identify caspase 3, 9 and PARP at once after stripping the same gel. Instead, the entire Western blot image was arranged in the order in which the same sample was used. In the case of the Western blotting, six tumors were used per group, and the manuscript was revised for this.
Round 2
Reviewer 1 Report
The authors did address some of the points raised within the first revision; however, there are some minor points that might need attention. Please find any remaining concerns regarding data presentation mentioned below as answers/questions (displayed in red) to the authors´ replies. Please note that the numbers (i.e. 9., 11, 14, 15, 16, 20) represent the numbers from the file of the first revision round. The reviewers adequately addressed any other points not mentioned below.
- Figure 1b: I encourage the authors to increase their n numbers, as the statistical analysis from 3 values might not be completely trustworthy. Hence, the statistics do not seem to match the presented data, i.e. a “*” significance for e.g. bifidum MG731 compared to the control seems incomprehensible considering the small error bar. Please provide the values of the individual measurements additionally to the bars and clarify the results obtained within the discussion.
Answer: We inserted individual measures into the graph and corrected the statistics.
Reviewer: Please clarify what was changed within the graphs, since the graphs seem to be the same as in the previous manuscript.
- Figure 3b,c,d: The authors MUST extend their description of the figures and method section. The description of Figure 3c is not sufficient to understand the data presented. Where all 8 tumors analyzed within Figure 3b and c or just some? Where all tumors analyzed in multiple measurements? Why was n=5 analyzed in Figure 3d and what happened to the other 3 mice per group (considering the authors´ statements that 8 mice per group were monitored)?
Answer: Of the 8 mice tested, 6 mice with normal tumor growth on average were selected per group and the data were used. Figures 3b,c,d used data from 6 tumors per group, respectively. We revised the incorrect part of the manuscript and added this part.
Reviewer: I appreciate that the authors corrected their description, however, the description is still not fully clear and the reviewers did not address all open questions. According to the authors’ statement within the “statistical analysis” part, they performed data analyzes in triplicates. Does that mean that they measured 3 independent experiments (biological replicates) for Figure 1 and 2 and performed those in triplicates (technical replicates)? According to the information within the methods section, one would assume that the authors analyzed six different mice also in triplicates, but the authors state that the 6 mice were measured once. So it seems that the statement within the emthods section that the data was analyzed in triplicates is nor correct.
- Figure 4a: It is unclear if the same sample was loaded twice or two different samples were loaded for the western blots. Was the same mouse tested in triplicates or were three samples from different mice analyzed? The authors stated that 8 animals were part of each group – where all 8 animals tested? The authors must clarify within the methods and the figure legends how many samples were analyzed (and used for the statistics) and if n=3 describes the triple analysis of the 8 animals or that only 3 animals were analyzed. If only 3 animals were analyzed, the authors should show and re-analyze the data of all animals.
Answer: Of the 8 mice tested, 6 mice with normal tumor growth on average were selected per group to use the data. We performed western blotting of all 6 tumors per group once. We corrected the errors in the manuscript.
Reviewer: If I understand this correctly, the authors analyzed 6 animals per group (i.e. 6 biological replicates), but only once (means 1 technical replicate). However, this is not stated in the paper and is also not in line with the statement within the methods section, where the authors described that all data was measured in triplicates. Please clarify and add appropriate information. Additionally, the authors did not address the comments regarding the samples analyzed within the two western blot lanes per gruoup. While it might be clear for the authors what was analyzed, it would be beneficial for the reader to state such information within the figure legend. Hence, the authors should state within their figure legends, if the same sample was loaded in duplicates or two different samples were loaded.
- Figure 4a: Considering the differences of lane 11 and 12 (3Mix-3), is seems that different samples were analyzed, as lane 12 displays more p53 with the same amount of protein loaded. Did the authors find any correlation with the tumor volume within these 2 samples? Similarly, within lane 9 and 10 (3Mix-1), lane 9 seems to display a more prominent p53 band compared to lane 10, although there seems to be slightly more protein loaded within lane 10. Besides physiological differences of live animals, one would assume a different tumor mass of this samples. Did the authors observe any correlation of the amount of e.g. p53 and the tumor volume? The authors should re-analyze their data accordingly and add a respective figure/panel showing any correlation.
Answer: Based on the reviewers' comments, we reanalyzed each tumor Western blot result.
Reviewer: The western blot images of the revised manuscript look a lot better than the images provided within the first manuscript. Just for clarification: did the authors also reanalyze all data (including the statistics)?
- Figure 4 b-e: In line with point 8, again, showing the individual values within the graphs would simplify the interpretation for the reader throughout all figures.
Answer: According to the reviewer's opinion, the labeling per group on the x-axis of the graph was reconfirmed.
Reviewer: This comment was not addressing the labeling of the axis. In fact, it was about showing the individually measured values as dots within the diagram additionally to the bars shown. This would give the reader additional information about the scattering of the individual values.
- Figure 6: The number of arrows within the images seems to be set voluntarily to visually match the western blot data. Can the authors describe how the position of the arrows was chosen? There are many dots that are not marked by arrows and arrows that do not seem to point to positive signals. The authors should remove all arrows, as they do not represent the actual results from the images. This would omit the false presentation of e.g. 3 arrows for p53 in MG5346, although the signals obtained is comparable to MG731 (where 7 arrows are shown). Interestingly, this arrow number “matches” the band intensity of the western blot data. Instead, the authors should show 2 representative arrows per image: one (e.g. in red) that displays the minimal signal that was counted as positive and one arrow (e.g. in blue) that points towards very clear signal. This would simplify the interpretation of the results. Additionally, I encourage the authors to re-analyze the images to determine the area of positive signal (in % of total tissue area shown, as it is hard to distinguish individual spots).
Answer: Immunohistochemistry is an experimental method for detecting a target protein in a tissue, and the expression level of the protein cannot be observed. Therefore, it is difficult to mark a signal that was considered positive or a clear signal, as the reviewers mentioned. We remarked with black arrows only the spots with 5 or fewer stained points in the image.
Reviewer: The authors should state in the methods or within the figure legend how they decided to set the arrows, as the arrows are not shown in every picture or every row.
Author Response
Reviewer 1 :
The authors did address some of the points raised within the first revision; however, there are some minor points that might need attention. Please find any remaining concerns regarding data presentation mentioned below as answers/questions (displayed in red) to the authors´ replies. Please note that the numbers (i.e. 9., 11, 14, 15, 16, 20) represent the numbers from the file of the first revision round. The reviewers adequately addressed any other points not mentioned below.
- Figure 1b: I encourage the authors to increase their n numbers, as the statistical analysis from 3 values might not be completely trustworthy. Hence, the statistics do not seem to match the presented data, i.e. a “*” significance for e.g. bifidumMG731 compared to the control seems incomprehensible considering the small error bar. Please provide the values of the individual measurements additionally to the bars and clarify the results obtained within the discussion.
Answer: We inserted individual measures into the graph and corrected the statistics.
Reviewer: Please clarify what was changed within the graphs, since the graphs seem to be the same as in the previous manuscript.
- Answer: All individual values were inserted in the graph, and it was confirmed that the statistical analysis was correct.
- Figure 3b,c,d: The authors MUST extend their description of the figures and method section. The description of Figure 3c is not sufficient to understand the data presented. Where all 8 tumors analyzed within Figure 3b and c or just some? Where all tumors analyzed in multiple measurements? Why was n=5 analyzed in Figure 3d and what happened to the other 3 mice per group (considering the authors´ statements that 8 mice per group were monitored)?
Answer: Of the 8 mice tested, 6 mice with normal tumor growth on average were selected per group and the data were used. Figures 3b,c,d used data from 6 tumors per group, respectively. We revised the incorrect part of the manuscript and added this part.
Reviewer: I appreciate that the authors corrected their description, however, the description is still not fully clear and the reviewers did not address all open questions. According to the authors’ statement within the “statistical analysis” part, they performed data analyzes in triplicates. Does that mean that they measured 3 independent experiments (biological replicates) for Figure 1 and 2 and performed those in triplicates (technical replicates)? According to the information within the methods section, one would assume that the authors analyzed six different mice also in triplicates, but the authors state that the 6 mice were measured once. So it seems that the statement within the methods section that the data was analyzed in triplicates is nor correct.
- Yes, as the reviewer mentioned, Figure 1 and 2 was generated from 3 independent experiments (biological replicates) and each experiment was performed technically in For the In vivo experiments, in fact, we injected tumor into 8 mice per group(therefore, 8 mice x 6 group = 48 mice), however, only, 75% of mice(36) generated a measurable size of tumor. Therefore, we randomly assigned 6 mice to each group. All in vivo experimental data were generated from 6 mice per group. We sincerely apologize for your confusion.
- Figure 4a: It is unclear if the same sample was loaded twice or two different samples were loaded for the western blots. Was the same mouse tested in triplicates or were three samples from different mice analyzed? The authors stated that 8 animals were part of each group – where all 8 animals tested? The authors must clarify within the methods and the figure legends how many samples were analyzed (and used for the statistics) and if n=3 describes the triple analysis of the 8 animals or that only 3 animals were analyzed. If only 3 animals were analyzed, the authors should show and re-analyze the data of all animals.
Answer: Of the 8 mice tested, 6 mice with normal tumor growth on average were selected per group to use the data. We performed western blotting of all 6 tumors per group once. We corrected the errors in the manuscript.
Reviewer: If I understand this correctly, the authors analyzed 6 animals per group (i.e. 6 biological replicates), but only once (means 1 technical replicate). However, this is not stated in the paper and is also not in line with the statement within the methods section, where the authors described that all data was measured in triplicates. Please clarify and add appropriate information. Additionally, the authors did not address the comments regarding the samples analyzed within the two western blot lanes per gruoup. While it might be clear for the authors what was analyzed, it would be beneficial for the reader to state such information within the figure legend. Hence, the authors should state within their figure legends, if the same sample was loaded in duplicates or two different samples were loaded.
- Answer: as the reviewer understood, we analyzed 6 animals per group (yes!! 6 biological replicates), but only once (yes!! 1 technical replicate).
And, Western blot images are two individual representative images out of six different tumor tissues, and the quantitative western blot results were analyzed from six different tumor tissue per group.
Figure 4a: Considering the differences of lane 11 and 12 (3Mix-3), is seems that different samples were analyzed, as lane 12 displays more p53 with the same amount of protein loaded. Did the authors find any correlation with the tumor volume within these 2 samples? Similarly, within lane 9 and 10 (3Mix-1), lane 9 seems to display a more prominent p53 band compared to lane 10, although there seems to be slightly more protein loaded within lane 10. Besides physiological differences of live animals, one would assume a different tumor mass of this samples. Did the authors observe any correlation of the amount of e.g. p53 and the tumor volume? The authors should re-analyze their data accordingly and add a respective figure/panel showing any correlation.
Answer: Based on the reviewers' comments, we reanalyzed each tumor Western blot result.
Reviewer: The western blot images of the revised manuscript look a lot better than the images provided within the first manuscript. Just for clarification: did the authors also reanalyze all data (including the statistics)?
- Answer: Yes, we also reanalyzed all data.
- Figure 4 b-e: In line with point 8, again, showing the individual values within the graphs would simplify the interpretation for the reader throughout all figures.
Answer: According to the reviewer's opinion, the labeling per group on the x-axis of the graph was reconfirmed.
Reviewer: This comment was not addressing the labeling of the axis. In fact, it was about showing the individually measured values as dots within the diagram additionally to the bars shown. This would give the reader additional information about the scattering of the individual values.
- Answer: as the reviewer recommended, All individual values were inserted in the graph.
- Figure 6: The number of arrows within the images seems to be set voluntarily to visually match the western blot data. Can the authors describe how the position of the arrows was chosen? There are many dots that are not marked by arrows and arrows that do not seem to point to positive signals. The authors should remove all arrows, as they do not represent the actual results from the images. This would omit the false presentation of e.g. 3 arrows for p53 in MG5346, although the signals obtained is comparable to MG731 (where 7 arrows are shown). Interestingly, this arrow number “matches” the band intensity of the western blot data. Instead, the authors should show 2 representative arrows per image: one (e.g. in red) that displays the minimal signal that was counted as positive and one arrow (e.g. in blue) that points towards very clear signal. This would simplify the interpretation of the results. Additionally, I encourage the authors to re-analyze the images to determine the area of positive signal (in % of total tissue area shown, as it is hard to distinguish individual spots).
Answer: Immunohistochemistry is an experimental method for detecting a target protein in a tissue, and the expression level of the protein cannot be observed. Therefore, it is difficult to mark a signal that was considered positive or a clear signal, as the reviewers mentioned. We remarked with black arrows only the spots with 5 or fewer stained points in the image.
Reviewer: The authors should state in the methods or within the figure legend how they decided to set the arrows, as the arrows are not shown in every picture or every row.
- Answer: It is true that the arrows are not shown in every picture as the reviewer observed, In fact, Brown dots in each IHC images present the expression of p-Akt, p53, Bax, cleaved caspase-9, -3 or PARP protein, individually. Most images show very clear brown dots(for example, p53 in MG5346 or 3Mix-3), however, some of dots are hard to identify(because of background staining) even though it clearly shows the expression of certain protein. We used black arrows to help readers so they can easily find the brown dots. Therefore, the amount of black arrows is not quantitatively correlated with the positivity of brown dots.
Thank you ,
Hyosun Cho